# Assessing Groundwater Mineralization Process, Quality, and Isotopic Recharge Origin in the Sahel Region in Africa

**Aboubacar Modibo Sidibé [1,2,3,*], Xueyu Lin [2] and Sidi Koné [3]**

[1] Laboratory of Groundwater and Environment, Jilin University, Changchun 130021, China
[2] College of Construction Engineering, Jilin University, Changchun 130021, China; xylin37@126.com
[3] National Directorate of Hydraulics, Square Patrice Lumumba BP 66, Mali; konesidi@hotmail.com
[*] Correspondence: aboubacar.sidibe@hotmail.fr; Tel.: +86-13019101204

**Abstract:** In the Sahel region in Africa, and in most arid regions, groundwater is the crucial source for water supply since surface water is scarce. This study aimed to understand a complex geochemical mechanism controlling the mineralization process in the Taoudeni Basin. A thousand randomly distributed groundwater samples acquired from different aquifers were used for this research. The results show that the majority of the samples observed are of the $Ca^{2+}$-$Mg^{2+}$-$HCO_3^-$ and $Na^+$-$HCO_3^-$ types depending on the different aquifers. $Mg^{2+}$ and $Ca^{2+}$ may react with $HCO_3^-$ precipitating as calcite and dolomite. The $Na^+$-$HCO_3^-$ groundwater type is mainly derived from the ion exchange process. This type indicates a paleo-marine depositional environment or that it passes through paleo-marine channels. Calcium of the standard $Ca^{2+}$-$HCO_3^-$ groundwater type exchanges with the sodium. Groundwater is characterized by the water-rock interactions that indicate the chemical alteration of the rock-forming minerals influencing its quality by a dissolution. The $\delta^2 H$ and $\delta^{18}O$ stable isotopes designate the evaporation importance in the basin and recharge with recent rain. The bicarbonate-type presence in groundwater suggests that it is young and fresh water. Multivariate statistical methods, notably Principal Component Analysis and Hierarchical Cluster Analysis, confirm affinities among the aquifers and identify three main clusters grouped into two water types. Cluster 1 consists of Infra-Cambrian and Quaternary aquifers, whereas cluster2 includes the Precambrian basement and Permian-Triassic aquifers.

**Keywords:** groundwater hydrochemical; isotopes; multivariate statistical analysis; Taoudeni Basin

---

## 1. Introduction

Groundwater is a vital livelihood resource for economic and social development. These waters are generally the primary source of drinking water for the population living in the Sahel-Sudanian region. The population growth rate is increasing by 3% per year affecting the groundwater quality and quantity demand. The degradation of groundwater quality is due to human activities and the lack of environmental protection policies. The groundwater reserves are significant, but the current climatic conditions at the study area, with annual rainfall between 600 and 1000 mm, do not allow a complete reconstitution of the water extracted by human activities.

Few studies have been conducted to understand groundwater quality and hydrochemical evolution in the study area, some of which include [1–3]. The previous reviews were focused mainly on water type determination in general. The present research also considers the mineralization process, the origins of groundwater quality, and the anthropogenic factors, in addition to the climate variability impact. In fact, climate variability affects groundwater fluctuations, and the rebound in

water levels can cause significant changes in water quality. In many aquifers, groundwater quality degradation is due to increasing salinity and is of anthropogenic origin. The pollution currently occurring increases further the stress on water resources by posing significant problems for ensuring water supply as well as maintaining natural ecosystems. This research incorporates both hydrochemical methods and a multivariate statistical approach to water quality monitoring for policymakers to address the ever-increasing water demands and water supply for different uses. The research method highlights the different chemical phenomena that can take place within these essential aquifers. The distinction between the different types of groundwater circulating in the regional multi-system aquifers is a complicated process to understand. To characterize groundwater, it is necessary to define the hydrochemical and isotopic criteria for discerning groundwater from a different origin [4]. Detailed groundwater quality and quantity estimations guarantee well-integrated water management systems. Understanding of the quality and evolution of groundwater hydrochemistry is crucial for the management and use of groundwater resources in the southern edge of the Taoudeni Basin.

The multivariate statistical methods offer many options for assessing groundwater types. These methods can be coupled with graphical representation to interpret the hydrochemical parameters. They are considered as the best and, often, the only practical solution for analyzing a large amount of hydrochemical data [5]. These techniques have been interpreted successfully by various studies that have implemented the hydrochemical process [6–8]. They identified the main factors controlling the groundwater quality in their respective study areas.

Cluster analysis has also been used prominently in water quality studies [7,9,10]. The cluster analysis, a multivariate classification procedure that detects natural groupings in the data under investigation, has been used widely alone or in conjunction with those defined by [11] and applied to various datasets.

Diagrams such as Piper and Schöeller can also be used to identify the chemical facies of water and relationships. Stable isotopes of oxygen and hydrogen are good indicators of groundwater recharge processes and evaporation [12–15]. In the context of groundwater compositions, Gibbs [16] proposed the principal natural mechanisms controlling global water chemistry, which can be classified as atmospheric precipitation, rock dominance, evaporation processes, and crystallization.

In view of climate change and anthropogenic pollution, the classification of groundwater quality and/or type can offer significant benefits, particularly in the regional management of groundwater [17–19]. Classification also makes it possible to follow the spatial evolution of the hydrochemical parameters and to estimate their mineral origin. Hydrogeological studies integrated with hydrochemical can be used to determine these factors impacts and the mechanisms controlling groundwater chemistry in the region. This research aimed to identify the mineralization and recharge processes as well as the origins of the groundwater and to assess its quality, focusing on the hydrochemical and isotopic data collected over three years (2014 to 2017).

## 2. Materials and Methods

### 2.1. Study Area

The study area is located at the Taoudeni Basin's southern edge, which is the primary geological formation and sedimentary deposit in West Africa. It covers a large part of the West African craton bordering Mauritania, Mali, and the southwestern part of Burkina Faso (Figure 1).

### 2.2. Geology and Hydrogeology Setting

#### 2.2.1. Geology of Study Area

The Taoudeni is the largest basin sedimentary syncline in the northwest part of Africa, formed in the middle of the Proterozoic (Figure 1). It continued to calm down until the middle of the Paleozoic when the deformation and uplift hercyniens took place. It contains up to 6000 m of sediments from

the Late Precambrian and the Paleozoic. Thin deposits of continental Mesozoic-Cenozoic, including the Dunes Quaternary and lake systems, cover it partially. These deposits show a spatial continuity between different basins, namely, the Taoudeni and the Tanezrouft Basins in the northern region and the Iullemeden Basins in the eastern region.

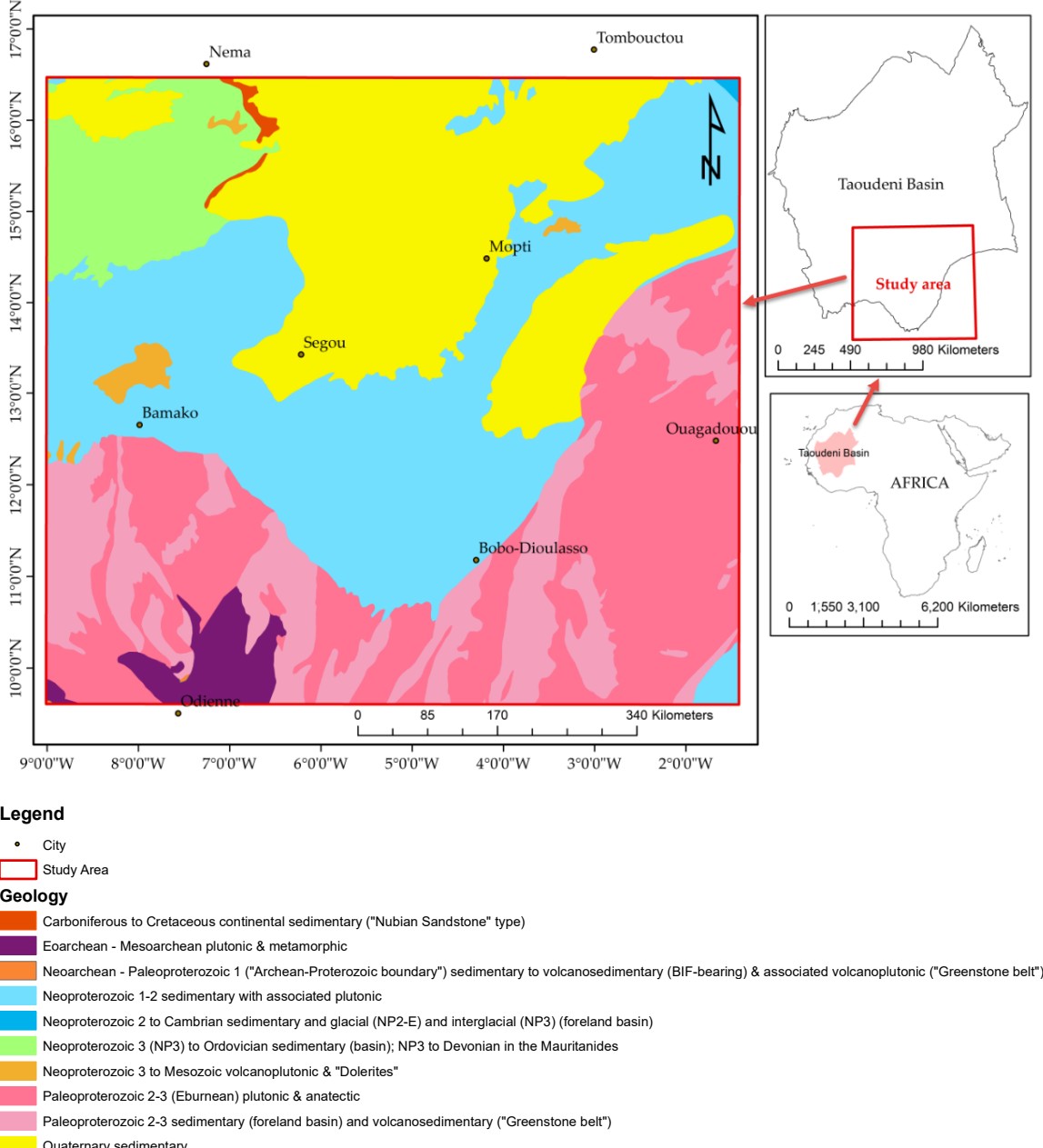

**Legend**

- • City
- ▭ Study Area

**Geology**
- ▮ Carboniferous to Cretaceous continental sedimentary ("Nubian Sandstone" type)
- ▮ Eoarchean - Mesoarchean plutonic & metamorphic
- ▮ Neoarchean - Paleoproterozoic 1 ("Archean-Proterozoic boundary") sedimentary to volcanosedimentary (BIF-bearing) & associated volcanoplutonic ("Greenstone belt")
- ▮ Neoproterozoic 1-2 sedimentary with associated plutonic
- ▮ Neoproterozoic 2 to Cambrian sedimentary and glacial (NP2-E) and interglacial (NP3) (foreland basin)
- ▮ Neoproterozoic 3 (NP3) to Ordovician sedimentary (basin); NP3 to Devonian in the Mauritanides
- ▮ Neoproterozoic 3 to Mesozoic volcanoplutonic & "Dolerites"
- ▮ Paleoproterozoic 2-3 (Eburnean) plutonic & anatectic
- ▮ Paleoproterozoic 2-3 sedimentary (foreland basin) and volcanosedimentary ("Greenstone belt")
- ▮ Quaternary sedimentary

**Figure 1.** Geological map of the Taoudeni Basin's southern edge.

## 2.2.2. Hydrogeology of the Study Area

Figure 2 shows the hydrogeology of the study area consisting of two principal aquifer systems according to the geological subdivision: the category of semi-continuous cracked aquifers (Neoproterozoic and Paleozoic) and the category of generalized aquifers (Quaternary). The semi-continuous or discontinuous fissured aquifer type, as a function of the density, the extent and the degree of intercalation of the crack networks, affects the host rock and the hydraulic relationships with the

layers in the cover. This aquifer type is usually found in sedimentary, crystalline, Precambrian, and primary formations.

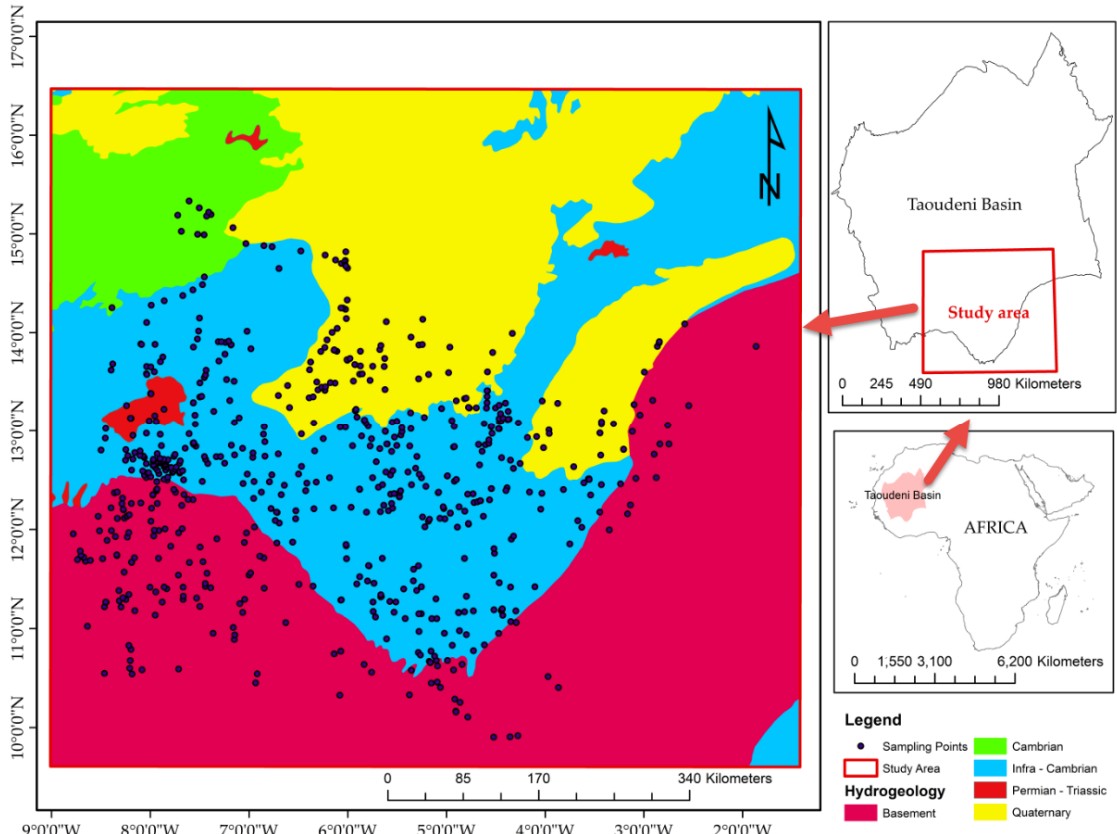

**Figure 2.** Hydrogeological map and sample location of the study area in the Taoudeni Basin's southern edge.

The tabular Infra-Cambrian outcrops (Neoproterozoic) mainly in the southern half of the study area contain the facies essentially sandstone and schists. The sandstone plateaus are composed of alternating sandstone benches. From the lithological point of view, the following succession is observed from bottom to top: the lower sandstone formation of the Sotuba Group with dolomitic sizes, and pelitic rock sequence intercalation, the Toun schists, the Koutiala sandstone, and the Bandiagara sandstone.

As for the Birimian Precambrian basement (Paleoproterozoic), it is flush with the south, southwest and west parts of the country and is the axial zone of the Iforas Adrar. In addition, the Precambrian basement is found in the extreme north of the study area, marking the northern limit of the Taoudeni Basin. The Birimian Precambrian basement (Paleoproterozoic) formations are either volcano-sedimentary or intrusive granitic. Thus, schists, greywackes, conglomerates, and quartzite, the facies of biotite granite, quartz diorites, and granodiorite, jasper, basalt, gabbro, dolerite, and tuff are observed. The generalized aquifer type is associated with the few consolidated formations with intergranular porosity, or none, encountered in the large sedimentary basins of the Secondary to Quaternary.

Quaternary aquifer outcrops in the study area. This aquifer is linked to the Niger River and contains the accompanying water table as it crosses the field of study. It covers the vast alluvial plains that extend on both sides of that river. The hydrogeological characteristics are related to the presence of permanent surface water and the extension of the flood zones covered by the floods in the Niger River. The average thickness of the continental formations is about 100 m. Continental deposits are often very clayey with depths ranging from 30 to 80 m.

## 2.3. Sampling and Analysis

A thousand hydrochemical and isotopic water samples collected on the Taoudeni Basin's southern border from 2014 to 2017 were used for this study (Figure 3). Their chemical and isotope analyses were carried out at the same time, and their analytical reports were collected from the National Directorate of Hydraulics in Mali. In 2016–2017, four hundred seventy-eight (478) samples were collected in Burkina Faso and Mali and analyzed at the Laboratory of Radio Analyses and Environment (LRAE, Tunisia), under the RAF/7/011 framework project in Mali and Burkina Faso [3]. In 2016–2017, one hundred and sixty-seven (167) samples acquired from another framework project called "PNMRE" [20] were also used for this research. In 2014–2015, one hundred and ten (110) samples from the PACTEA framework project [21] were as well included for this research. All 267 samples collected from Mali alone were analyzed at the National Laboratory of Water of Mali. The water samples collected from the southern edge of the Taoudeni Basin (Mali and Burkina Faso), mainly from boreholes, were analyzed in compliance with international standards. The sample points were randomly chosen depending on the aquifer type and frequency of use as a groundwater source by the population from the cities and/or villages. These samples were collected to enable an analysis of the groundwater mineralization processes and the origins of water pollution. Thirteen (13) hydrochemical parameters and two isotopic parameters were chosen for this study. The hydrochemical parameters are alkalinity, Electric Conductivity (EC), potential hydrogen (pH), Total Dissolved Solids (TDS), calcium ($Ca^{2+}$), magnesium ($Mg^{2+}$), chloride ($Cl^-$), nitrate ($NO_3^-$), Sulphur dioxide ($SO_4^{2-}$), sodium ($Na^+$), potassium ($K^+$), bicarbonate ($HCO_3^-$), and silicon oxide ($SiO_2$). Some samples were collected at different times between 2014 and 2017 from every sample point. This was to determine the different hydrochemical parameters at each point.

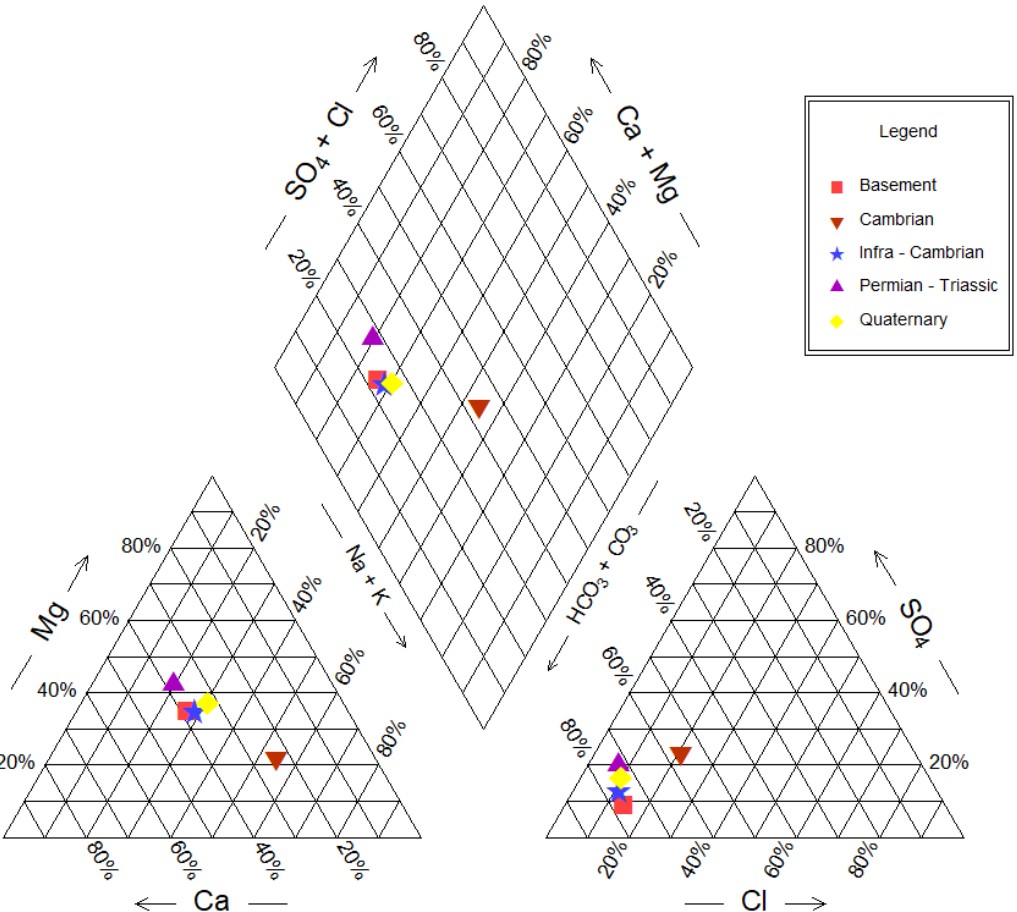

**Figure 3.** Piper diagram showing the study area's groundwater type.

*2.4. Graphical Data Processing*

The software "Rock Ware AQ.QA" was used to determine the groundwater type of different aquifers in the study area. The Piper and Schöeller diagrams were used to show the nature of the relationship between the groups, using the different hydrochemical settings. The Gibbs diagrams were utilized to enhance understanding of the hydrochemical process on the groundwater's chemistry in the study area. The oxygen and hydrogen stable isotopes were tested at the Laboratory of Radio Analyses and Environment (LRAE, Tunisia) under the RAF/07/011 framework project. The measures were standardized according to reports on the standard water average of Vienna (‰ vs. Smow). These stable isotopes were used to determine groundwater origins.

*2.5. Multivariate Statistical Data Processing*

The Minitab 18.1 and STATISTICA software were used to analyze the groundwater chemistry data in this study. Principal Component Analysis (PCA) [15,22,23] was applied to extract the main factors controlling the groundwater chemistry of each aquifer type and reduce the secondary factors. The rotation of the principal components (PCs) was performed using the varimax method. Factor Analysis (FA) highlights the critical factors responsible for the variation of groundwater quality. Factor analysis was implemented to extract the most significant elements and reduce the contribution of the less significant variables. These methods, which are generally used to examine the relationships among variables, are frequently used for environmental research with a high degree of temporal or spatial variation. Statistical approaches allow the conversion of geochemical data into an easily interpretable form (called factors).

Hierarchical Cluster Analysis (HCA) is a statistical approach that was used for grouping the groundwater sampling sites by aquifer type, clustering them together, and studying the hydrochemical parameters according to their origins. The results are displayed in a dendrogram of distance that is considered the best method to display the hierarchical cluster analysis results [24,25].

## 3. Results

*3.1. General Characteristics of Groundwater*

The general hydrochemical statistics for all the borehole samples taken from the study area are shown in Table 1 to understand their status. The statistics determine their trend evolution among the different aquifers in the Taoudeni Basin's southern edge.

**Table 1.** Descriptive statistics of the hydrochemical parameters. Alkalinity; Electric Conductivity (EC,) potential hydrogen (pH); Total Dissolved Solids (TDS), calcium ($Ca^{2+}$); magnesium ($Mg^{2+}$); sodium ($Na^+$); potassium ($K^+$); chloride ($Cl^-$); Sulphur dioxide ($SO_4^{2-}$); nitrate ($NO_3^-$); bicarbonate ($HCO_3^-$); silicon oxide ($SiO_2$).

| Variable | Mean | Standard Deviation | Coefficient of Variation | Minimum | Median | Maximum | Range |
|---|---|---|---|---|---|---|---|
| Alkalinity | 52.67 | 83.46 | 158.45 | 0.16 | 13.05 | 397 | 396.84 |
| EC | 263.9 | 314.7 | 119.23 | 2.5 | 177.5 | 2660 | 2657.5 |
| pH | 6.93 | 0.93 | 13.47 | 3.6 | 7.13 | 9.35 | 5.75 |
| TDS | 220.5 | 252.8 | 114.64 | 5 | 149.3 | 1988.2 | 1983.2 |
| $Ca^{2+}$ | 18.57 | 22.18 | 119.42 | 0.33 | 11.49 | 251.02 | 250.69 |
| $Mg^{2+}$ | 11.1 | 13.49 | 121.46 | 0.146 | 6.58 | 116.96 | 116.96 |
| $Na^+$ | 13.08 | 27.4 | 209.48 | 0.02 | 5.2 | 358.56 | 358.56 |
| $K^+$ | 8.32 | 53.36 | 641.48 | 0.09 | 3.78 | 1392 | 1391.91 |
| $Cl^-$ | 8.27 | 19.45 | 235.19 | 0.1 | 4.24 | 315.08 | 315.08 |
| $SO_4^{2-}$ | 11.38 | 39.26 | 344.95 | 0.1 | 2 | 458.72 | 458.72 |
| $NO_3^-$ | 17.68 | 68.83 | 389.26 | 0.1 | 2.25 | 926.62 | 926.62 |
| $HCO_3^-$ | 105.11 | 91.98 | 87.5 | 1 | 83.98 | 460 | 459 |
| $SiO_2$ | 11.95 | 8.61 | 72.08 | 2 | 9.19 | 51.1 | 49.1 |

For this study area, the pH level is varying, ranging from 3.6 to 9.35 confirming that groundwater is moderately acidic to alkaline. Some rock and soil types, such as limestone, can neutralize acid more effectively than other types of rock and soil, such as granite. Human activities can also affect the pH of nearby water sources. TDS is generally made up of inorganic salts and amounts of small organic matter with a maximum value of 1988 mg/L, $NO_3^-$ value has a maximum of 926 mg/L in some samples that confirm that there are human activities taking place in the area under study. Potassium has a maximum value of 1392 mg/L in some areas that are unacceptable for human consumption. The $K^+$ come from orthoclase and muscovite of granite, and its high concentration in some groundwaters seems to be an anthropogenic factor. The proportions of $Na^+$, $Cl^-$, and $SO_4^{2-}$ are increasing, indicating their anthropogenic origin. Sodium, occurring at levels higher than 200 mg/L in combination with chloride, gives water a salty taste. $Na^+$ in the study area has a maximum value of 358 mg/L in some aquifers. Most likely, these minerals come from mineral weathering in some rocks as well as human activities. The agriculture and urban runoff can carry excess minerals into water sources, the same occurring with wastewater discharges and industrial wastewater.

Water type is dependent on factors such as the aquifer's lithological characteristics and the groundwater's flow retention time and flow pattern. To distinguish the different water types in the studied aquifer system, major ions were plotted on a Piper tri-linear diagram [26]. This diagram reveals that 90% of groundwater samples do not contain any dominant cation (Figure 3). The triangular cation diagram reveals that groundwater chemistry is divided into two groups. The first group consists of $Ca^{2+}$-$Mg^{2+}$ waters found in the majority of samples from the southern edge of the Taoudeni Basin. The second group contains $Na^+$ groundwater type, similar to the type of groundwater found in deserts. $Na^+$-$HCO_3^-$ water type forms when fresh groundwater invades an area that previously contained seawater or a sodium-rich brine derived from seawater. In some places, mostly in Cambrian aquifer areas, the $Na^+$-$HCO_3^-$ type indicates a paleo-marine depositional environment or that it passes through paleo-marine channels. The spatial distribution of these chemical facies is linked essentially to the same lithological nature of aquifer and recharging conditions. Overall, mineralization and major ionic structure are not discernible, suggesting that there is a flow of desert to adjacent areas [16,26–28]. However, in the case of anions, the samples are of bicarbonate type, considering the hydrochemical facies according to Kehew [29]. $HCO_3^-$ also dominates in all the study area's groundwater, although some other anions are larger locally ($SO_4^{2-}$ or $Cl^-$). The average trend of cations and anions in the study area is as follows: $Ca^{2+} > Na^+ > Mg^{2+} > K^+$ and $HCO_3^- > NO_3^- > SO_4^{2-} > Cl^-$, respectively.

The Schöeller diagram (Figure 4) is also used in this research to determine the water quality type and assist in distinguishing similar patterns in the anion and cation ratios. Water concentrations are a function of the groundwater hydrochemistry and aquifer's rocks' chemical composition. According to the Schöeller diagram, the majority of water samples are distinguished into the $Ca^{2+}$-$Mg^{2+}$-$HCO_3^-$ group and the $Na^+$-$HCO_3^-$ group. This result obtained from the Schöeller diagram seems to be compatible with the results acquired from the Piper diagram.

To understand the different associations between the main hydrochemical variables, a Pearson correlation matrix (Table 2) was established to determine the relationships between the different variables or chemical descriptors. This matrix shows a very strong positive correlation between TDS and EC; $Ca^{2+}$, $Mg^{2+}$, $NO_3^-$, $Cl^-$, $Na^+$, and $SO_4^{2-}$; $Mg^{2+}$ and $HCO_3^-$; EC and $Mg^{2+}$; EC and $Ca^{2+}$; $Ca^{2+}$ and $NO_3^-$; $Mg^{2+}$ and Alkalinity; $Mg^{2+}$ and $HCO_3^-$. A moderate uphill (positive) relationship is observed between $Na^+$ and $SO_4^{2-}$; EC and $Cl^-$; $Ca^{2+}$ and $Mg^{2+}$; $Cl^-$ and $NO_3^-$; EC and $HCO_3^-$; $NO_3^-$ and EC; $Na^+$ and $Cl^-$. This means that paired parameters have a strong to moderate influence between them.

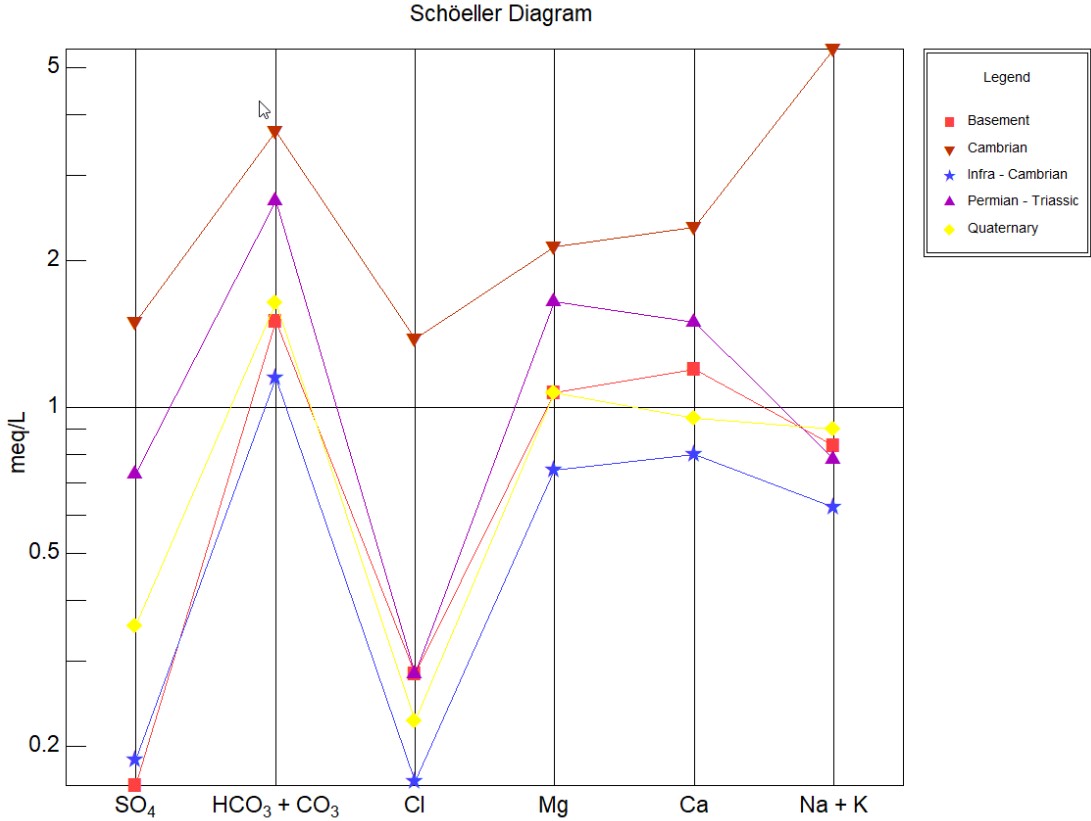

**Figure 4.** Schöeller diagram distinguishing similar patterns in the anion and cation ratios.

**Table 2.** Pearson correlation matrix.

|  | Alkalinity | EC | pH | TDS | $Ca^{2+}$ | $Mg^{2+}$ | $Na^+$ | $K^+$ | $Cl^-$ | $SO_4{}^{2-}$ | $NO_3{}^-$ | $HCO_3{}^-$ |
|---|---|---|---|---|---|---|---|---|---|---|---|---|
| EC | 0.57 | | | | | | | | | | | |
| pH | 0.07 | 0.36 | | | | | | | | | | |
| TDS | 0.52 | 0.85 | 0.34 | | | | | | | | | |
| $Ca^{2+}$ | 0.43 | 0.73 | 0.31 | 0.84 | | | | | | | | |
| $Mg^{2+}$ | 0.71 | 0.78 | 0.32 | 0.84 | 0.66 | | | | | | | |
| $Na^+$ | 0.29 | 0.6 | 0.29 | 0.73 | 0.33 | 0.41 | | | | | | |
| $K^+$ | 0.16 | 0.07 | 0.01 | 0.11 | 0.11 | 0.06 | 0.05 | | | | | |
| $Cl^-$ | 0.21 | 0.68 | 0.16 | 0.76 | 0.56 | 0.58 | 0.5 | 0.1 | | | | |
| $SO_4{}^{2-}$ | 0.34 | 0.49 | 0.19 | 0.7 | 0.36 | 0.56 | 0.69 | 0.06 | 0.6 | | | |
| $NO_3{}^-$ | 0.27 | 0.62 | 0.04 | 0.77 | 0.73 | 0.51 | 0.42 | 0.13 | 0.64 | 0.35 | | |
| $HCO_3{}^-$ | 0.63 | 0.65 | 0.54 | 0.61 | 0.56 | 0.73 | 0.44 | 0.03 | 0.23 | 0.31 | 0.1 | |
| $SiO_2$ | 0.33 | 0.45 | 0.37 | 0.52 | 0.39 | 0.48 | 0.49 | 0.11 | 0.35 | 0.46 | 0.3 | 0.54 |

*3.2. Hydrochemical Process*

Complex geochemical mechanisms and processes mainly control the study area's groundwater. The dissolved ion concentration in groundwater samples is frequently governed by lithology, the nature of geochemical reactions, and the solubility of interaction rocks. The Gibbs graph [16] was used to understand the relationship between the composition of water and its respective aquifer characteristics according to the variation in the ratio of Na/(Na + Ca) and Cl/(Cl + HCO₃) as a function of TDS. Figure 5 shows that the majority of samples in the study area are characterized by rock-water interactions, which are an indicator of the chemical alteration of rock-forming minerals influencing the quality of groundwater by rock dissolution through which the water flows beneath the surface. The cations and anions are derived mainly from rock weathering rather than evaporation, crystallization, and precipitation. The proportion of major ions can be derived from the crystalline dolomitic limestone and

calcium–magnesium silicates weathering, chiefly from calcite, gypsum, and feldspar plagioclase [28]. Cambrian waters are predominantly evaporative due to semi-arid climate causes, conditions, or sources of surface contamination.

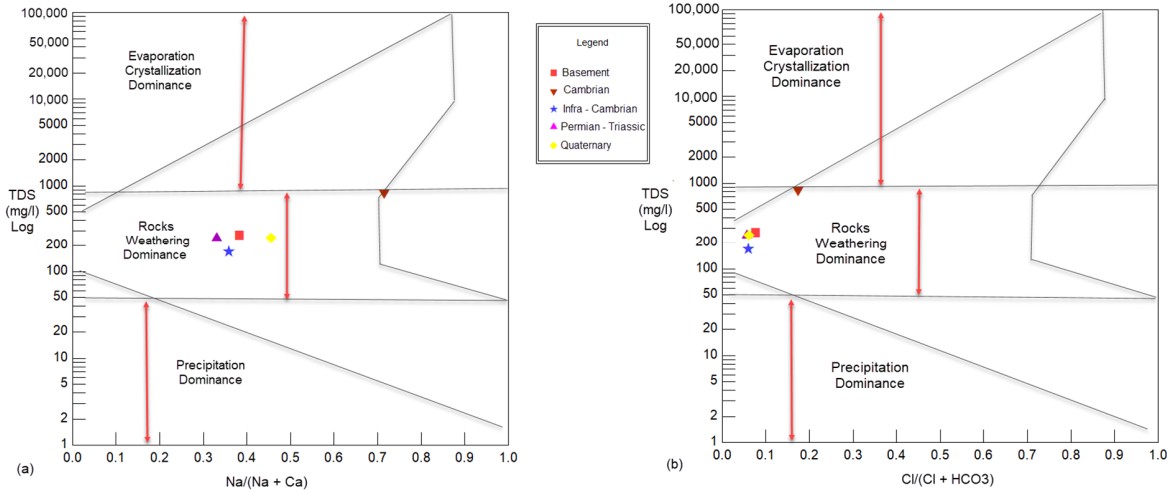

**Figure 5.** Mechanisms governing groundwater chemistry according to a Gibbs diagram: (**a**) TDS vs. $Na^+/(Na^+ + Ca^{2+})$; (**b**) TDS vs. $Cl^-/(Cl^- + HCO_3^-)$.

The $Mg^{2+}$ ion occurs because of chemical weathering and dissolution of dolomite, marls, and other rocks. The $Mg^{2+}$ ion is seldom dominant in natural waters, which is also true for the Taoudeni Basin's southern edge. Since carbonates are present in many different rock types, including most sedimentary rocks, and even some igneous and metamorphic rocks, carbonate chemistry is relevant to the evolution of most groundwater. The calcite main dissolution mechanism is represented by Equation (1):

$$CaCO_3 + CO_2\ (g) + H_2O = Ca^{2+} + 2HCO_3^-. \tag{1}$$

This reaction includes the following step:

$$CO_2\ (g) + H_2O = H^+ + HCO_3^-, \tag{2}$$

which is the reaction of carbon dioxide with water, to produce the hydrogen ions (acidic conditions) that promote the dissolution of calcite by the following reaction:

$$CaCO_3 + H^+ = Ca^{2+} + HCO_3^-. \tag{3}$$

From this reaction, we can see that the calcite solubility is controlled by the amount of carbon dioxide available; the more the $CO_2$, the more calcite will dissolve. The calcite solubility is dependent on pH; therefore, the lower the pH (more hydrogen ions expected), the more calcite will dissolve. Other processes such as oxidation of sulfide minerals or reactions of sulfur pollutants in the air can also produce hydrogen ions that will promote the dissolution of calcite.

$Na^+$, $Ca^{2+}$, and $HCO_3^-$ can be derived mainly from the weathering of plagioclase feldspar that is important to dominant minerals in most igneous rocks, and many metamorphic rocks. $HCO_3^-$ is primarily associated with precipitation and subsurface lithological characters (particularly limestone aquifer). Low $HCO_3^-$ in Infra-Cambrian aquifers indicates low degrees of water-rock interactions in the form of weathering $Na^+$ and $SiO_2$. $Na^+$ and $Cl^-$ are the factors that lead to an increase in salinity due to evaporation mainly in the Cambrian aquifer. Enrichment in $Na^+$ is related to water-rock interaction processes (volcanic rocks being rich in sodium). The $Na^+$ is due to either NaCl remnants or perhaps from sodic feldspar and feldspathoids contributed in the country rock/sediment. $Ca^{2+}$ ions are successively replacing the $Na^+$, adsorbed to the clay surfaces in order of increasing affinity for the

clay or due to weathering of plagioclase feldspar in the groundwater. The primary source mineral of $Mg^{2+}$ is biotite and chlorite of granite. The silicate dissolution may be a probable sodium source in the study area's groundwater. The primary sources of silica are from silicate rocks composed of quartz, chert, feldspars, and clay minerals. $SiO_2$ presence may also depend on various dissociation and dissolution processes during rock-water interaction and contributions from anthropogenic sources. $Na^+$ and $SiO_2$ are likely to be from weathering of sodic feldspar. The plagioclase feldspar is usually the first primary mineral to be weathered significantly, followed by K-feldspar and biotite at a later stage, whereas quartz should be the most stable of all the minerals. Given the short residence time of shallow groundwater, K-feldspar and biotite may not be weathered significantly.

### 3.3. Stable Hydrogen and Oxygen Isotope Signature

Deuterium and oxygen-18 stable isotopes provide essential information on water origin, recharge under different climatic conditions and mixing of waters from various sources [30]. Results from environmental isotope data indicate that groundwater in the study area has spatial variable isotope levels. Thus, the $\delta^{18}O$ values range from −7.98 to +2.1‰ (vs. Smow) and the deuterium content obtained varies between −51.52 and −6.49‰ (vs. Smow). The correlation between the $\delta^2H$ and $\delta^{18}O$ values of the water samples in the study region is shown in Figure 6, and the grouped samples are plotted near the global meteorological water line (GMWL). Samples plotted below the GMWL [31] indicate relatively higher evaporation before infiltration. Generally, the value of $\delta^2H$ and $\delta^{18}O$ decreases progressively as the elevation of the recharge mountain increases. The $\delta^{18}O$ and $\delta^2H$ values sampled in this study are plotted and compared to the local meteorological water line (LMWL: $\delta^2H = 7.2\ \delta^{18}O + 12$) and the global meteorological water line (GMWL: $\delta^2H = 8\delta^{18}O + 10$) (Figure 6).

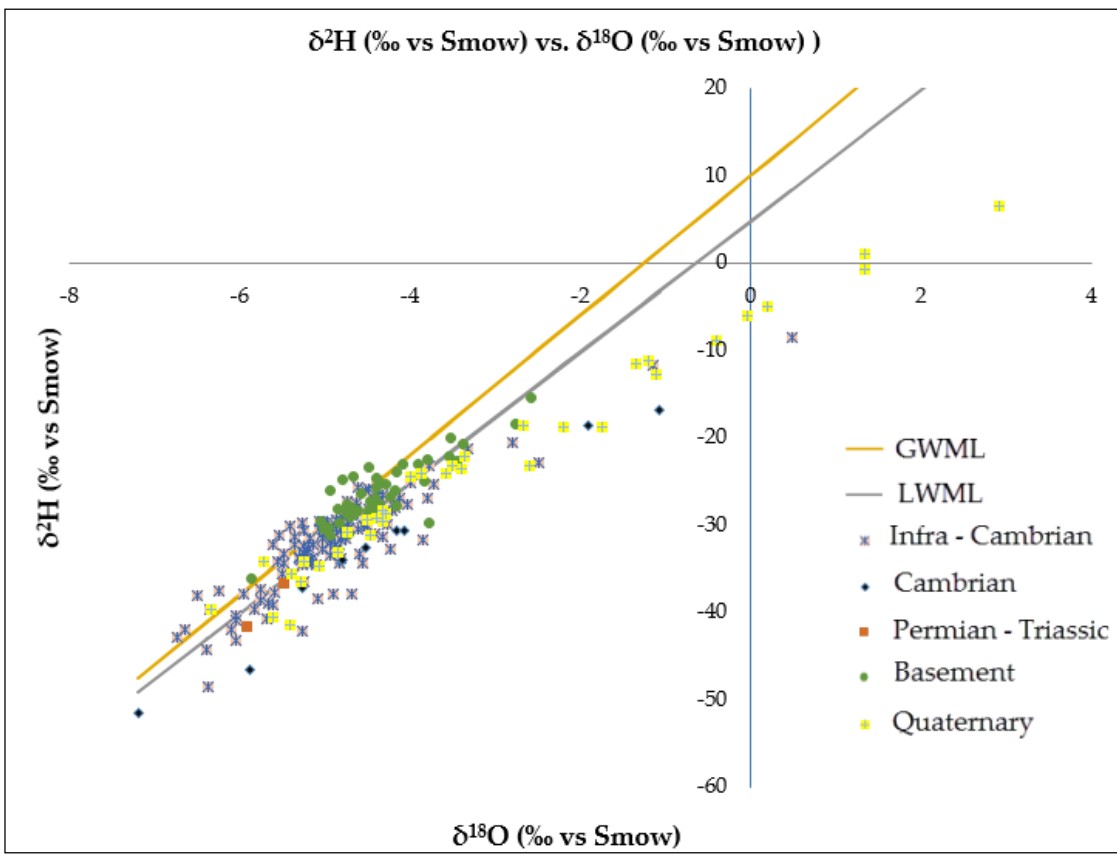

**Figure 6.** $\delta^2H$ (‰ vs. Smow) vs. $\delta^{18}O$ (‰ vs. Smow).

The comparison of the median values of Oxygen-18 and deuterium for each group shows a low variability of 1‰ ($\delta^{18}O$) and 8‰ ($\delta^{2}H$), and these values are consistent with the weighted isotopic amount of annual precipitation. High variability is observed in the Quaternary aquifer revealing the evaporation process as indicated by enriched values. This evaporation could occur before infiltration (endorheic local area collecting surface runoff) or during penetration according to the conditions of the soil helping ability, direct penetration from the upper water level near the surface (or directly from an open well). Many wells in the Quaternary formation are shallow, and the depth/isotope chart confirms this process.

It is noted that groundwater in the study area is lying on the GMWL and therefore has isotope signatures close to those of the meteoric waters that contribute to their recharge. They are groundwater infero flux (alluvial layers of underflow) which are in the beds of wadis located at particular areas of recharge.

### 3.4. Multivariate Statistical Analysis

This subsection presents results about both bottom-up HCA and PCA synthesized with appropriate statistical software.

### 3.4.1. Hierarchical Cluster Analysis

In this study, the water type classification is performed by incorporating HCA and constructing a dendrogram. The resulting dendrogram suggests three groups according to their degree of similarity (Figure 7). The groups are chemically distinct from one another, and the Euclidean distance measures their similarities. The waters of the same group have a small connection distance between them according to their resemblance. Groundwaters from the Infra-Cambrian and Quaternary compose cluster 1, and those from Precambrian basement and Permian-Triassic, cluster 3. The Cambrian aquifer is constituted mainly of the cluster 3. To describe the characteristics of each group of samples, Table 1 represents the median values of the hydrochemical and isotopic data used in the hierarchical group analysis.

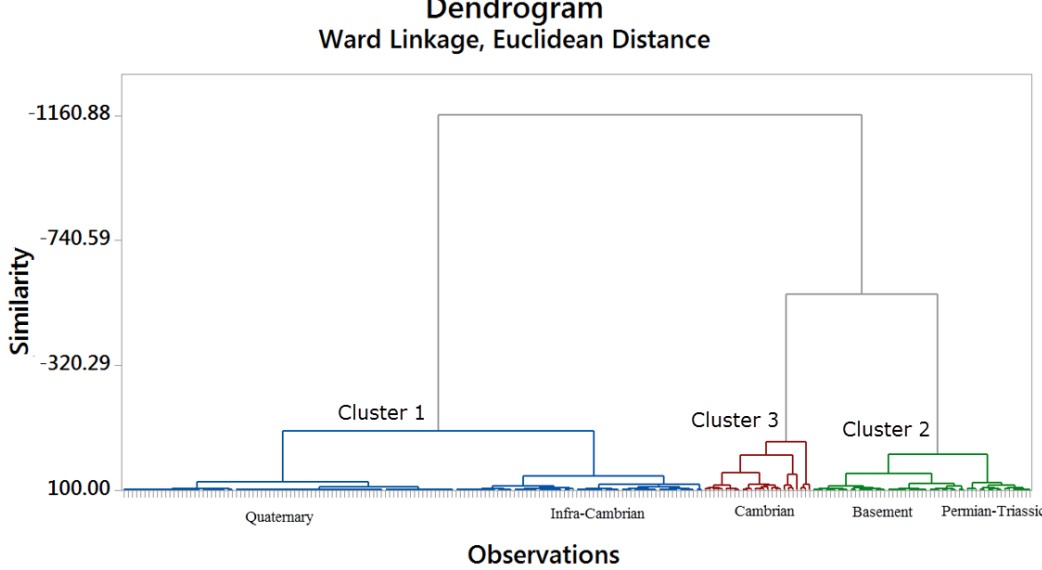

**Figure 7.** Dendrogram of the observations of the study area.

The distance dendrogram of the hydrochemical parameters in the Taoudeni Basin's southern edge is presented in Figure 7. The final cluster classification (Figure 8) is obtained from the cluster analysis based on the distance of the correlation coefficients of the parameters studied. The relationship between the chemical elements is examined through the dendrogram of the 15 chemical parameters. According to the observations of the dendrogram, the variables are grouped into three clusters (Figure 9).

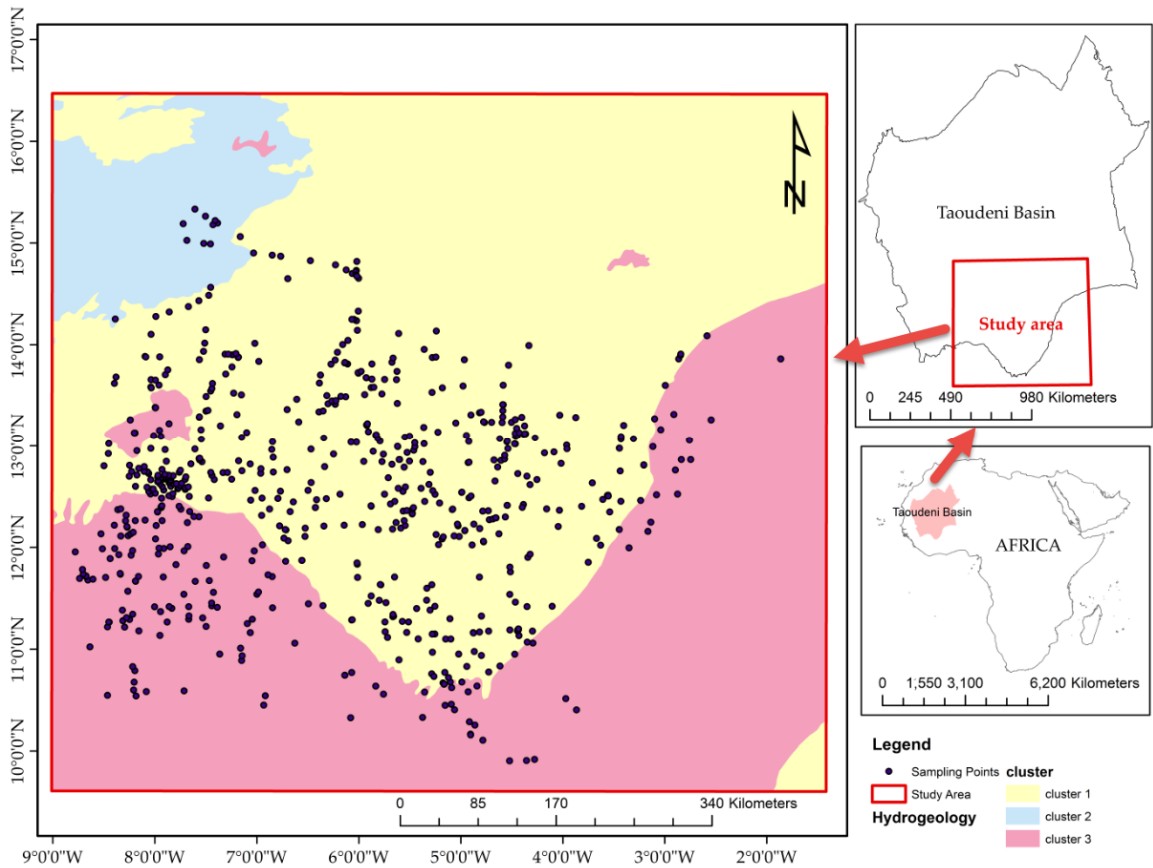

**Figure 8.** Cluster map.

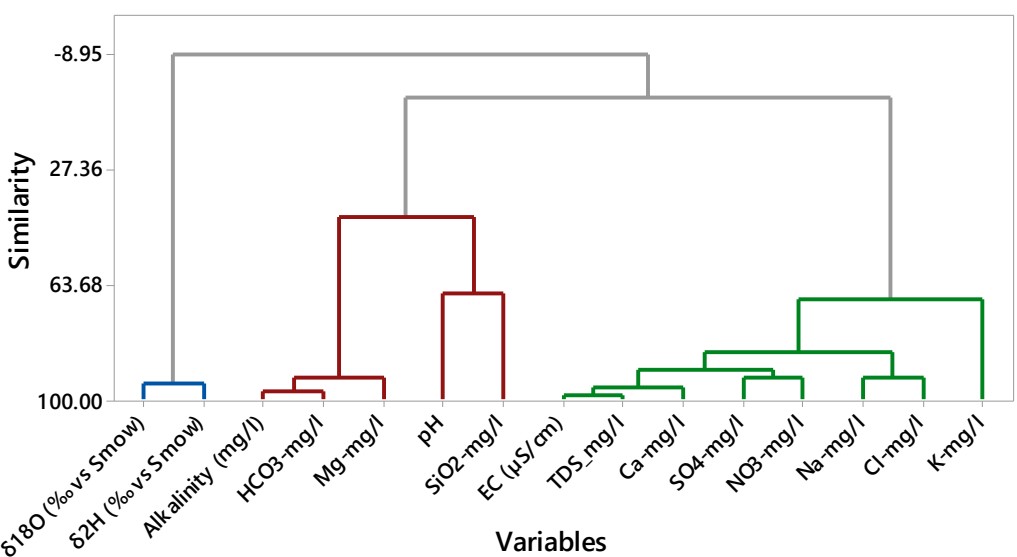

**Figure 9.** Dendrogram of the variables in the study area.

Cluster 1 consists of Quaternary and Infra-Cambrian aquifers. The primary element shows mainly lower content in Infra-Cambrian and Quaternary. Many observations from the surface water deviate from the LMWL due to fractionation by evaporation. The evaporation process marks the Infra-Cambrian and Shallow Quaternary aquifer showing a higher range with more depleted materials.

Observations from these shallow aquifers (Figure 6) is coinciding with trend lines, probably originating from a point corresponding to the average composition in precipitation.

Cluster 2 comprises the waters of Permian-Triassic and Precambrian basement. They are defined by alkalinity, $Mg^{2+}$, $HCO_3^-$, and $SiO_2$. The bicarbonate is the predominant anion in the Infra-Cambrian groundwater and shows lower values due to lower EC. In groundwater, the dissolved $Mg^{2+}$ is predominant. The natural balances that control magnesium levels in groundwater are complex. Many cation exchanges, adsorption, and desorption reactions (on clay minerals) influence the solution of magnesium in the underground environment. In igneous rocks, volcanic rocks, and weathering rocks containing clay, the magnesium solution is more difficult than in carbonate rocks. The magnesium shows a lower value in Infra-Cambrian and Precambrian basement aquifers.

Cluster 3 is composed of Cambrian waters and high concentrations of EC, TDS, $Ca^{2+}$, $Na^+$, $K^+$, $Cl^-$, $SO_4^{2-}$ and $NO_3^-$. In the Cambrian aquifer, almost all values are high, and this geological formation has a high content in sodium linked to bicarbonate and also to chloride. The sulfate contents are high; anions prevailing in this water, in this case, are associated with a high content of sodium, which may be linked to evaporitic minerals locally (thenardite or glauberite). Nitrate ions in the Sahel region are naturally occurring in the soil with specific vegetation like acacias, which is a nitrogen-fixing tree. Termites also can produce nitrogen, and as a result, groundwater can show high nitrate contents.

### 3.4.2. Principal Component Analysis (PCA)

A particular problem in water quality monitoring is the complexity associated with analyzing a large number of measured variables [32]. Therefore, in this study, groundwater quality data were grouped using FA. The variable correlation matrix was generated, and factors were extracted by the centroid method, rotated by varimax. From the results of the FA, the first three eigenvalues were found to be bigger than 1 (Table 3). According to Table 3, a subsequent interpretation of the factor loadings, the first three components were extracted, and the other components were eliminated.

**Table 3.** Eigenvalues and inertia.

| Value | Eigenvalues Extraction: Principal Components | | | |
| --- | --- | --- | --- | --- |
| | Eigenvalue | % Total (variance) | Cumulative (Eigenvalue) | Cumulative (%) |
| 1 | 8.273532 | 55.15688 | 8.27353 | 55.15688 |
| 2 | 2.341451 | 15.60967 | 10.61498 | 70.76655 |
| 3 | 1.562250 | 10.41500 | 12.17723 | 81.18155 |

The first three factors reflect most of the information sought and make it possible to represent the point cloud significantly because the sum of the variance expressed by these factors is 81.18% of the variance in data sets. The factor F1 (55.16%), the factor F2 (15.61%), and the factor F3 (10.41%) account for most of the total variance of the original dataset (Figure 10). The rotation was established for the variables of the three principal components to interpret factorial axes that were easy to understand. Table 4 represents the factorial weights of the three principal components obtained after rotation. For each element, we only considered values which were more significant than 0.7. The loading of three factors parameter on two from FA is associated with each factor station, well defined and contributes slightly to other factors. This helps not only in the interpretation of the results but also in the identification of anthropogenic sources of pollution from the groundwater quality data.

Each factor is defined by some essential variables in the demonstration of the groundwater mineralization mechanism. The first factor (F1) is high, and positively correlated with EC, TDS, $Ca^+$, $K^+$; $Cl^-$, $SO_4^{2-}$, $Na^+$, and $NO_3^-$ representing the mineralization, but in a relationship with contamination of nitrate and sulfate in groundwater. The hydrochemical variable $Ca^{2+}$ originates at first glance from mineralization of the geological components of soils as well as from a moderate decrease of pH concentration. The $SO_4^{2-}$ levels may be due to agricultural activities [33], and the contribution of $Na^+$ to this factor can be considered to be a result of cation-exchange processes in the

soil–water interface [4]. The sources of dissolved $SO_4^{2-}$ in groundwater may include dissolution of sedimentary sulfate, oxidation of both sulfide minerals and organic materials, and anthropogenic inputs.

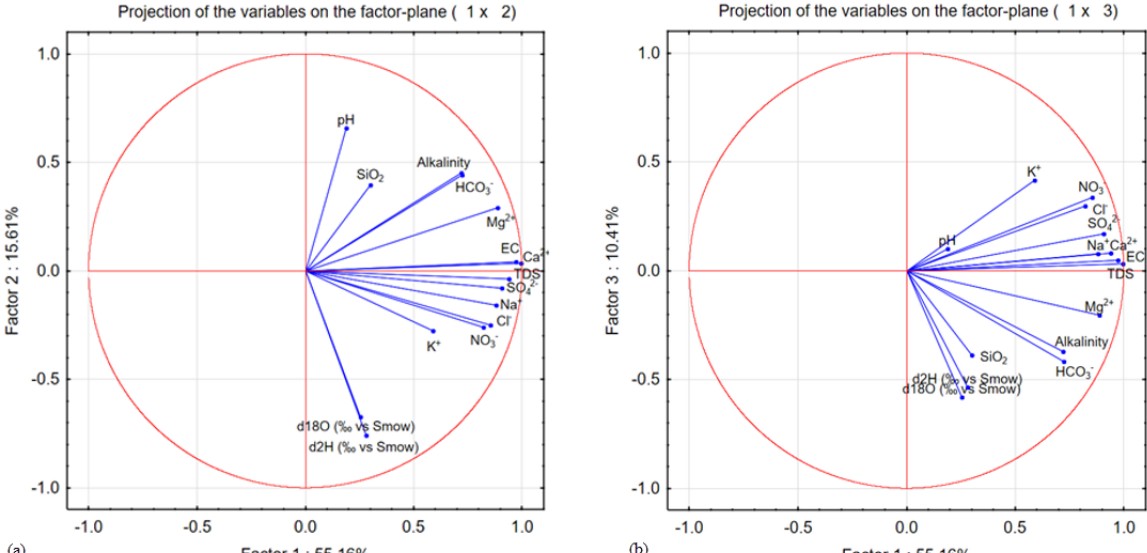

**Figure 10.** Space of the variables of the factorial plane: (**a**) F1-F2; (**b**) F1-F3.

**Table 4.** Factorial weights of principal main components (With rotation).

| Variable | Factor Loadings (Varimax Raw) Extraction: Principal Components (Marked Loadings Are > 0.700000) | | |
|---|---|---|---|
| | Factor (1) | Factor (2) | Factor (3) |
| $\delta^{18}O$ (‰ vs. Smow) | 0.104068 | 0.917797 | 0.077950 |
| $\delta^{2}H$ (‰ vs. Smow) | 0.171700 | 0.956457 | 0.015448 |
| Alkalinity | 0.315608 | −0.007108 | 0.874186 |
| EC | 0.831634 | 0.069975 | 0.505878 |
| pH | 0.040812 | −0.535040 | 0.433144 |
| TDS | 0.842157 | 0.088729 | 0.521957 |
| $Ca^{2+}$ | 0.839073 | 0.102251 | 0.419566 |
| $Na^{+}$ | 0.819606 | 0.188765 | 0.318467 |
| $Mg^{2+}$ | 0.578335 | 0.030601 | 0.760625 |
| $K^{+}$ | 0.762966 | 0.025085 | −0.112112 |
| $Cl^{-}$ | 0.899772 | 0.118096 | 0.094088 |
| $SO_4^{2-}$ | 0.864434 | 0.076224 | 0.322779 |
| $NO_3^{-}$ | 0.942556 | 0.090697 | 0.094161 |
| $HCO_3^{-}$ | 0.299246 | 0.030762 | 0.895932 |
| $SiO_2$ | −0.031617 | −0.010778 | 0.627712 |
| Explained Variance | 6.370608 | 2.132948 | 3.673677 |
| Prp. Totl | 0.424707 | 0.142197 | 0.244912 |

The second factor (F2) expressing evaporites represents stable water isotopes. The third factor (F3) represents the contribution hydrochemistry of the groundwater. The $Mg^{2+}$ is an elemental metal, which increases the alkalinity of the environment [22].

The factorial plane F1-F2 (Figure 11) reveals three poles of mineralization and highlights three classes of the aquifer in the study area. Two groups of samples can be identified separately for each axis according to parameters F1 and F2. F1 shows more mineralized samples, F2 has samples that show the most enriched isotopic values and therefore an excess of lower deuterium, and F3 represents the axis of hydrochemistry waters.

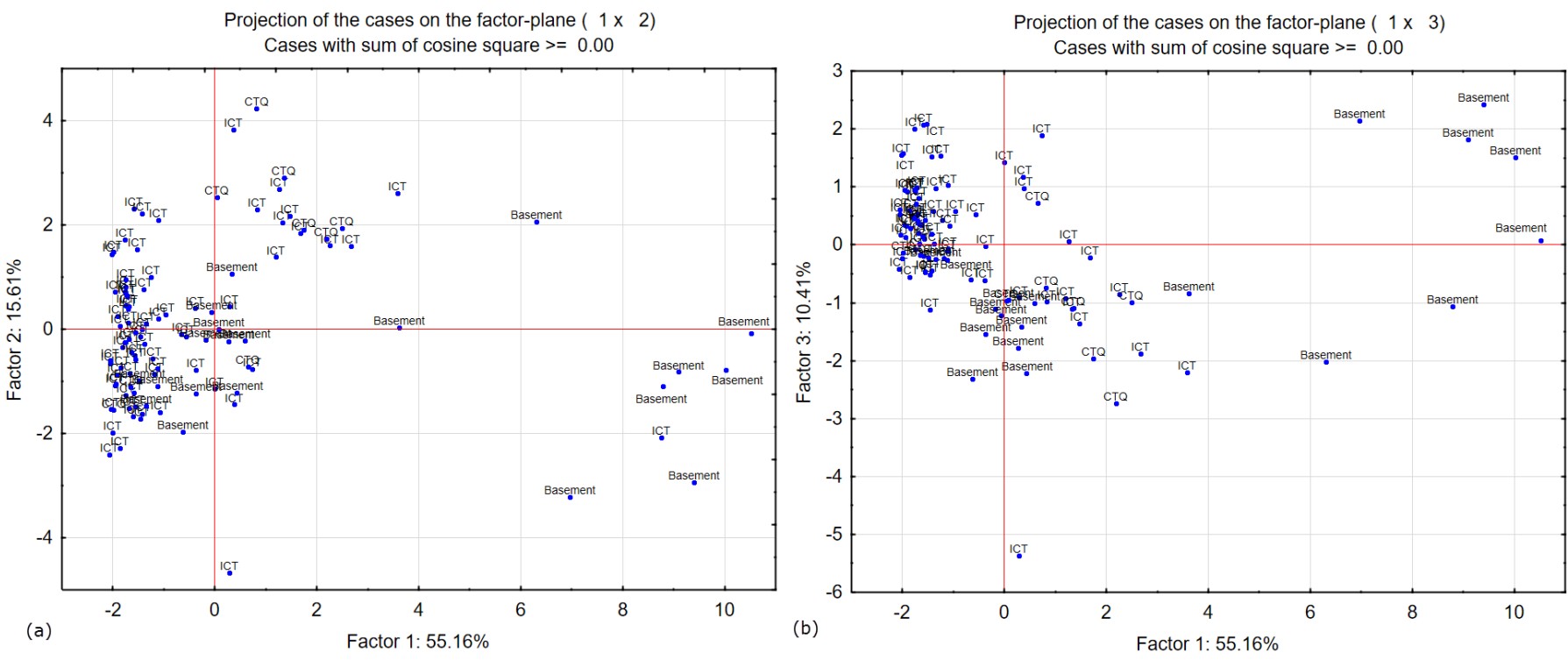

**Figure 11.** Space of the statistical units of the factorial plane: (**a**) F1-F2; (**b**) F1-F3.

## 4. Discussion

This study incorporates hydrochemical, isotopic, and multivariate statistical methods to examine groundwater quality, endeavoring to understand the origin of the mineralization process and origin. Groundwater quality is controlled primarily by the dominance of the inter-water exchange rocks with some evaporated samples. The results indicate that groundwater in the southern edge of the Taoudeni Basin is generally alkaline, moderately saline, and heavily loaded in the Cambrian and to a lesser extent in the Permian-Triassic and Precambrian basement.

The Cambrian waters are sodium bicarbonate whose presence in natural waters can be attributed to the dissolution of the halite. Those of Infra-Cambrian, Quaternary, Precambrian basement, and Permian-Triassic are calcium bicarbonate and magnesium. The bicarbonate may be due to the demise of calcite $CaCO_3$ found to be present throughout the Sahel region. The $Ca^{2+}$ ion is derived from the alteration of calcium-containing minerals such as aluminum silicates that are present in rock magmatic. Magnesium can be caused by the hydrolysis of ferromagnesian minerals, which can undergo base exchange phenomena in clay levels.

There is an increase in salinity, sulfate, and nitrate, which is probably due to anthropogenic causes. Their concentrations are considerably high in some aquifers and can pose some problems related to consumption and domestic purposes. The calcium and magnesium bicarbonate types (Ca-HCO$_3$ type) are identified in the Quaternary, Infra-Cambrian, Precambrian basement, and Permian-Triassic aquifers. As for the Cambrian aquifer, the water contains sodium bicarbonate (Na-HCO$_3$ type) that indicates paleo-marine dispositional environment or that it passes through paleo-marine channels. Calcium of the standard $Ca^{2+}$-HCO$_3^-$ groundwater type exchanges with the sodium (Na$^+$).

The Gibbs diagram suggests that the chemistry of water is mainly regulated by the dissolution of the minerals forming the rock. The results demonstrate that groundwater recharge in the study area is entirely through precipitation. The shallow groundwater of the continental Quaternary aquifer terminates evaporation, and the evaporation line is above the GMWL. The isotopic composition of groundwater samples from aquifers of the Infra-Cambrian fracture and isotope-depleted crystalline Precambrian basement suggests the presence of infiltrated groundwater during wintering.

The stable isotope $\delta^{18}$O and $\delta^2$H values of the measured groundwater are scattered around the LMWL, indicating that the water is of meteoric origin. The grouping of observed groundwater samples suggests that evaporation and isotopic exchange with aquifer minerals may occur in the system. Evaporation is more marked in shallow Quaternary, and there are some samples of Infra-Cambrian aquifers due to the proximity of the piezometric level to the soil surface in these aquifers. It is also noted that many wells are open in Quaternary aquifers. The disposition of some samples below the LMWL is depleted due to rain during transit or isotopic exchange with aquifer materials that are less in $\delta^{18}$O. The majority of Infra-Cambrian and Precambrian basement samples are found above LMWL, suggesting a recent recharge history, but they may have evaporation effects, resulting in some fractionation, leading to the enrichment of surface water. It is noted that recharge occurs mainly in Infra-Cambrian and Precambrian basement aquifers by significant fractures.

The PCA and HCA suggest that anthropogenic pollutants and all natural soluble levels explain most of the variations. The main problems affecting groundwater quality in the southern edge of the Taoudeni Basin are salinity and point pollutants, particularly in the Cambrian aquifer; this can lead to a deterioration of groundwater quality in the surrounding environment.

## 5. Conclusions

Taoudeni edge southern Basin in Africa is considering as an important agriculture, industry, livestock, and population areas. This research aims to determine the mineralization process, recharge condition, and quality origin of groundwater. The principal economic role of the aquifers is to provide high-quality drinking water for communities and towns as well as to supply water for agricultural and industrial use.

The methodology implemented in this paper allows a better evaluation of the different sources of groundwater and of the hydrochemical and anthropogenic processes impacting on the groundwater's chemistry. The results obtained can be used as a fundamental basis for transboundary water management in the Taoudeni Basin (1,500,000 km$^2$), which is one of the biggest shared sedimentary basins in Africa. Most of the studies in the region focus on the pure determination of the water type in highly localized aquifers. The approach employed in this study gives relevant information according to the aquifer settings in the large Taoudeni Basin with a considerable number of samples. Hydrochemical classical analysis methods confirmed by multivariate statistics distinguish two main water types, namely, $Ca^{2+}$-$Mg^{2+}$-$HCO_3^-$ and $Na^+$-$HCO_3^-$. In particular, water-rock interactions indicate that chemical alteration of the rock-forming minerals influences the quality of the region's groundwater.

Moreover, these natural processes, the return flow of irrigation waters taking place within the region relating to agricultural practices, and the artificial recharge areas play a significant role in the salinization of the study area's groundwater. The highest mineralization is observed in the Cambrian aquifer in general. The stable isotope results show a significant recharge by recent rainwater and its highest variability is found in the Quaternary aquifer revealing the evaporation process as indicated by enriched values. A high level of contamination by nitrate and also by potassium, chloride, sulfate, and sodium is observed in the region.

Indeed, water availability in some regions is currently being affected severely by anthropogenic factors and climate change. Groundwaters' origin and their mineralization process determination are an essential aspect of hydrogeological investigations which allow the establishment of drinking water protection zones in the region. Furthermore, these investigations will help determine the possible geogenic or anthropogenic contamination potential and its impact on the extracted groundwater.

The information provided by this research may be useful in groundwater's quality management. It can also be used as an essential tool in groundwater monitoring to assist planners, decision-makers, managers, and local officials by supplying water for different uses. Policymakers can also use hydrochemical characteristics to develop a water management strategy according to ever-increasing water demands for different purposes.

We recommend further research on other aquifers within the Taoudeni Basin to confirm the general mineralization process mechanism and groundwater quality.

**Author Contributions:** S.K. performed all the hydrochemical and isotopic data collection. A.M.S. did the analysis and report writing under the supervision of X.L.

**Funding:** This research received no external funding.

**Acknowledgments:** Field research would not have been possible without the assistance of the National Directorate of Hydraulics of Mali. The authors sincerely thank the Chinese government through the Chinese Scholarship Council for supporting the thesis research. The University of Jilin provided us with expertise and an ideal framework for conducting studies, analysis, and research.

**Conflicts of Interest:** The authors declare no conflict of interest.

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
