# Peer review of "Assessing Groundwater Mineralization Process, Quality, and Isotopic Recharge Origin in the Sahel Region in Africa"

_water, doi:10.3390/w11040789_

Round 1
Reviewer 1 Report
Notes to consider when improving work: The research was carried out in 2014-2017. How was the location of the sampling points selected? Were the samples taken at a single time at selected points, were the tests repeated? From what points were the water collected (wells, boreholes, water intakes)? Were the water collected exposed to pollution (landfills, sewage disposal)?
Analysis of test results:
The pH is a negative logarithm from the concentration of hydrogen ions. You can`t pull out averages or calculate statistics based on an average.
Author Response
Response to Reviewer 1 Comments
Notes to consider when improving work:
Point 1: The research was carried out in 2014-2017. How was the location of the sampling points selected? Were the samples taken at a single time at selected points, were the tests repeated? From what points were the water collected (wells, boreholes, water intakes)? Were the water gathered exposed to pollution (landfills, sewage disposal)?
Response 1: The borehole water sampling is done randomly according to the aquifer setting in the study area, and we consider among all data those have a chronicle of measurement. The hydrochemical and isotopic test in series were done in National Laboratory of Water in Mali for some points and others in the Laboratory of Radio Analyses and Environment (LRAE, Tunisia), under the RAF/7/011 framework project in Mali and Burkina. We are witnessing anthropogenic pollution on almost all of the study area as the highly populated area practicing intensive agricultural activity. The gold mines and above all the traditional gold panning in recent years has become the new pollution factor with heavy metals uses.
Point 2: Analysis of test results:
The pH is a negative logarithm from the concentration of hydrogen ions. You can`t pull out averages or calculate statistics based on an average.
Response 2: The pH is a measure of the hydrogen ion concentration of the water as ranked on a scale of 1.0 to 14.0. The statistics on the pH are done in general content of groundwater in the study area to figure up its value range from 3.6 minimum to 9.35 maximum that in our case allows confirming that groundwater is slightly acidic to alkaline.

Reviewer 2 Report
Comments to the Authors
The manuscript describes some very interesting to examine the results of hydrochemical methods using multivariate statistics. The results are also quite novel since they show significant influence for water resource quality management. However, the manuscript is not publishable in its present state. Some of the concerns are listed below. If the authors correct the manuscripts, it can be accepted.
1. Figure 1 should be changed clearer.
2. In page 398, authors mention “develop integrated water….”. Can authors explain how to integrate water resource management.
3. Authors should check the references format carefully. There are some errors in references chapter. For example [16], R. j. should be changed to R. J..
4. In your entire article, any elements should be corrected, such as Ca2+ should be Ca2+, etc…
5. In chapter 2.3, authors should mention the era of data collected and analyzed.
Author Response
Response to Reviewer 1 Comments
Comments to the Authors
The manuscript describes some very interesting to examine the results of hydrochemical methods using multivariate statistics. The results are also quite novel since they show significant influence on water resource quality management. However, the manuscript is not publishable in its present state. Some of the concerns are listed below. If the authors correct the manuscripts, it can be accepted.
Point 1: Figure 1 should be changed clearer.
Response 1: Done
Point 2: In page 398, authors mention “develop integrated water….”. Can authors explain how to integrate water resource management.
Response 2: The development of an integrated water management strategy is a process that promotes the coordinated management of water within the defined area according to water ever-increasing demands for different uses. It allows optimizing, in a fair way, the socio-economic well-being that is resulting without compromising the sustainability of vital ecosystems since the water resources are becoming scarcer. The management strategies integrate the methods considering beyond hydrogeologic analyses using a strict disciplinary focus which can be employed to assess the factors of aquifer management or policy defined by both science and consensus condition.
Point 3: Authors should check the references format carefully. There are some errors in references chapter. For example [16], R. j. should be changed to R. J..
Response 3: Done
Point 4: In your entire article, any elements should be corrected, such as Ca2+ should be Ca2+, etc…
Response 4: Done
Point 5: In chapter 2.3, authors should mention the era of data collected and analyzed.
Response 5: Done

Reviewer 3 Report
Manuscript concerns issue of the assessing groundwater mineralization process. The form of the manuscript is typical case study concerning the Sahel region. Article should be carefully edited according to Author’s guidelines, references, etc. References, tables and figures have to follow instruction for author and should have the similar type of font as the rest of the text. The title should be more concise Assessing groundwater mineralization process, quality and isotopic recharge origin in the Sahel region under climate variability effect and pollution: Case base on hydrochemical methods and multivariate statistics. Include in the introduction and the conclusion the value added with respect to existing research The quality of the figures should be improved, the font size is too small, eg. Line 204: Figure 6. Gibbs Diagram: (a) TDS vs Na+/ (Na+ + Ca2+), (b) TDS vs Cl-/ Cl- + HCO3-). It is really hard to read the font of this figure. Please check the rest of the diagrams. Line 70: What about data from 2018? In the following section 3.1. General characteristics of groundwater: line 159, add some introduction before table, as The groundwater’s general characteristics in the study area are shown in Table 1. Line 191: Table 2. Pearson correlation matrix, in calculation of Pearson correlation there is lack of dot: 0.57. Check in the whole text the superscript of the parameters as in Table 2. You show in two different way: Ca2+ and Ca2+. Line 157-158. You don’t need to write it: This section is divided by subheadings to present concise and precise description of the experimental results, their interpretation as well as the preliminary conclusions that can be drawn. Number the formula/reaction. Line 154: Wrong word order: The results are in the dendrogram displayed, please check the whole text. Line 269. Figure 8. Dendrogram of the observations of the study area presents only 5 observations? All samples should be presented not mean values. Why You chose method of complete linkage not Ward's method in the cluster analysis? Line 274, style: According to the observation of the dendrogram, the variables are into three clusters grouped. What is the reason of grouping variables? 298: Figure 10. Dendrogram of the variables in the study area. Line 396: This research is an essential tool in groundwater monitoring that can be useful to assist planners, decision-makers, managers and local officials in water supply for different uses. What implementation (uses?) do You mean, in more detail. The authors provided little results and discussion of the performed calculations. They just inform to readers the obtained results. I recommend to describe obtained results in more precisely way, what would reinforce the considered problem.
Author Response
Response to Reviewer 3 Comments
Point 1: Manuscript concerns issue of the assessing groundwater mineralization process. The form of the manuscript is typical case study concerning the Sahel region. Article should be carefully edited according to Author’s guidelines, references, etc. References, tables and figures have to follow instruction for author and should have the similar type of font as the rest of the text. The title should be more concise Assessing groundwater mineralization process, quality and isotopic recharge origin in the Sahel region under climate variability effect and pollution: Case base on hydrochemical methods and multivariate statistics.
Include in the introduction and the conclusion the value added with respect to existing research
Response 1: Done
Point 2: The quality of the figures should be improved, the font size is too small, eg. Line 204: Figure 6. Gibbs Diagram: (a) TDS vs Na+/ (Na+ + Ca2+), (b) TDS vs Cl-/ Cl- + HCO3-). It is really hard to read the font of this figure. Please check the rest of the diagrams.
Response 2: Done
Point 3: Line 70: What about data from 2018?
Response 3: The research data come from two different countries since this study is not supported Data collection and fieldwork are difficult. The uses data were collected in the same
Point 4: In the following section 3.1. General characteristics of groundwater: line 159, add some introduction before table, as The groundwater’s general characteristics in the study area are shown in Table 1.
Response 4: Done
Point 4: Line 191: Table 2. Pearson correlation matrix, in calculation of Pearson correlation there is lack of dot: 0.57. Check in the whole text the superscript of the parameters as in Table 2.
Response 4: The correction is done in Table 2, and all the document contain
Point 5: You show in two different way: Ca2+ and Ca2+.
Response 5: Correction is done
Point 5: Line 157-158. You don’t need to write it: This section is divided by subheadings to present concise and precise description of the experimental results, their interpretation as well as the preliminary conclusions that can be drawn. Number the formula/reaction.
Response 6: Done
Point 7: Line 154: Wrong word order: The results are in the dendrogram displayed, please check the whole text.
Response 7: Done
Point 8: Line 269. Figure 8. Dendrogram of the observations of the study area presents only 5 observations? All samples should be presented not mean values. Why You chose method of complete linkage not Ward's method in the cluster analysis?
Response 8: For this research, we used about boreholes 1000 samples analysis and the five observation represent the aquifer’s setting in the study area. To highlight the figure8 we use only 5 observation, and the figure8 are changed according to all samples observation in the area.
The complete linkage method defines the distance between two groups as the distance between their two farthest-apart members. The ward method usually yields clusters that are well separated and compact. With this method, groups are formed so that the pooled within-group sum of squares is minimized. That is, at each step, the two clusters are fused which result in the least increase in the pooled within-group sum of squares. The recommendation was relevant, and the correction is done in this section.
Point 9: Line 274, style: According to the observation of the dendrogram, the variables are into three clusters grouped. What is the reason of grouping variables? 298: Figure 10. Dendrogram of the variables in the study area.
Response 9: The observation dendrogram is used to define the relationship between the water type and the variable dendrogram defined the relation between each chemical element.
Point 10: Line 396: This research is an essential tool in groundwater monitoring that can be useful to assist planners, decision-makers, managers and local officials in water supply for different uses. What implementation (uses?) do You mean, in more detail. The authors provided little results and discussion of the performed calculations. They just inform to readers the obtained results. I recommend to describe obtained results in more precisely way, what would reinforce the considered problem.
Response 10: Integration of allocation and monitoring strategy that promotes the coordinated management of water within the defined area according to water ever-increasing demands for drinking, industries, and agriculture. These strategies will allows optimizing, in a fair way, the socio-economic well-being that is resulting without compromising the sustainability of vital ecosystems since the water resources become scarce.
Corrections were made to this section according to the recommendations.

Round 2
Reviewer 3 Report
The manuscript concerns important topic of analysis of the hydrochemical characteristics.
The title of the manuscript should be more concise.
Line 744. What software You used? In the manuscript there is lack of reference to the Figure 8. Dendrogram of the variables in the study area line: 789, so I understand it was deleted. Some errors concerning references occur in the lines 817-818: found to be bigger than 1 (Error! Reference source not found.). According to Error! Reference source not found, also in line 954: factorial plane F1-F2 (Error! Reference source not found.). Please check the obtained similarities in the Figure 6. Dendrogram of the observations of the study area line 767, as they reached really high values. Some information in the conclusions are repetition of the previous section of obtained results. State in the conclusions how your findings differ or support those of other studies and why, also provide a synthesis of arguments presented in the paper to show how these converge to address the research problem and the overall objectives of your study. What were the unique or new contributions your study made to the overall research about the examined issue?
Some of the future perspectives and information about implementation in practice were deleted (why? It is so important):
The results provide some information that may be
useful for water resource quality management. This
research is an essential tool in groundwater monitoring
that can be useful to assist planners, decision-makers,
managers and local officials in water supply for
different uses. Policymakers can also use hydrochemical
characteristics to develop integrated water resources
management policies.
But it is really important to highlight the need for further research, which will provide the reader with evidence that the manuscript covers an in-depth awareness of the research problem.
Author Response
The document has been corrected according to the recommendations made. Minitab18 and STATISTICA software are used for statistical studies. We use 1000 randomly selected samples from different aquifers in the study area, which is why observations reach high values.
